# Damage Sensitive Signals for the Assessment of the Conditions of Wind Turbine Rotor Blades Using Electromagnetic Waves

Zainab Riyadh Shaker Al-Yasiri [1,†] ⓘ, Hayder Majid Mutashar [1,*], Klaus Gürlebeck [2] and Tom Lahmer [1] ⓘ

1  Institute of Structural Mechanics-Structural Analysis and Component Strength, Bauhaus-Universität Weimar, Marienstrasse 15, 99423 Weimar, Germany
2  Institute of Mathematics/Physics, Coudraystr 13B, Bauhaus-Universität Weimar, 99423 Weimar, Germany
*  Correspondence: hayder.majid.mutashar@gmail.com
†  Current address: Al-Nahrain University, Baghdad 64074, Iraq.

**Abstract:** One of the most important renewable energy technologies used nowadays are wind power turbines. In this paper, we are interested in identifying the operating status of wind turbines, especially rotor blades, by means of multiphysical models. It is a state-of-the-art technology to test mechanical structures with ultrasonic-based methods. However, due to the density and the required high resolution, the testing is performed with high-frequency waves, which cannot penetrate the structure in depth. Therefore, there is a need to adopt techniques in the fields of multiphysical model-based inversion schemes or data-driven structural health monitoring. Before investing effort in the development of such approaches, further insights and approaches are necessary to make the techniques applicable to structures such as wind power plants (blades). Among the expected developments, further accelerations of the so-called "forward codes" for a more efficient implementation of the wave equation could be envisaged. Here, we employ electromagnetic waves for the early detection of cracks. Because in many practical situations, it is not possible to apply techniques from tomography (characterized by multiple sources and sensor pairs), we focus here on the question of whether the existence of cracks can be determined by using only one source for the sent waves.

**Keywords:** wind turbine rotor blades; electromagnetic waves; crack detection; Empire XPU 8.01; Matlab

## 1. Introduction

For many years now, Europe has been enjoying the expansion of renewable energy technologies, with wind turbines (wind power energy) being one of them, with innumerable onshore and offshore turbines [1–3]. In comparison to fossil fuels such as oil, coal, and natural gas, which all have a limited supply source, renewable energy sources will never be used up and are also much more environmentally friendly. Currently, renewable energy is the best alternative to getting energy and, above all, clean energy.

In the meantime, the lifespan of the first systems has been reached, and the replacement and/or dismantling of these systems is taking place. Since the rotor blades are specially made of hybrid materials, aluminum, epoxy, carbon fibers, or the latest generation of fiberglass, their recycling, in particular with hybrid materials, is still difficult and costly, see references, e.g., [4–6]. Therefore, early removal should be avoided. To support the decision, continuous monitoring of the blade is necessary and sufficient to know about the existence of cracks. The shape and location of the cracks are less important as local repair activities seem unrealistic. Furthermore, due to practical and economic restrictions, the monitoring should be based on a simple arrangement of the devices for the measurements. Ref. [7] presents a comprehensive analysis of the state of the art in wind turbine blade design.

Due to cost savings and weight reduction, two-bladed wind turbines may become increasingly popular. Using 3D printing technology, many rotors with varying numbers

of blades and comparable performance behaviour were developed and built in [8]. Additionally, in [9], they examine existing literature on how a wind turbine's number of blades influences its effectiveness and ability to generate electricity.

In [10], they created a novel active damping technology for wind turbines to minimize wind turbine tower vibrations. Moreover, due to the high dynamic forces, structural changes (cracking) may occur in the majority of fragile materials even during their lifetime. As a result, continuous or periodic inspection for the identification of possible faults is a major issue today, and powerful, reliable, and inexpensive testing (inspection) techniques should be created and implemented.

Non-destructive testing (NDT) techniques are commonly used in industry, especially in the aeronautic domain, to evaluate the properties of a wide variety of materials without causing damage. For the first time, we will use this technique for the detection and characterization of flaws or mechanical/physical properties in the blades of wind turbines. Several techniques are used in the NDT field, including ultrasonic testing, electromagnetic testing [11], thermography testing, radiographic testing, liquid penetration testing, and magnetic particle testing. In [12,13], a review is given of the main NDT techniques used for different materials.

EMW-NDT is a novel non-destructive testing approach based on electromagnetic wave (EMW) technology and has been proposed in [14]. The EMW-NDT approach has been shown to be useful in detecting damage such as cracks or other defects.

In [15], a sensing concept based on guided electromagnetic waves (GEW) is presented to detect material defects such as delaminations, cracks, or inclusions.

Recently, electromagnetic waves and guided electromagnetic waves have been introduced as a novel approach for monitoring surface damage of metallic structures, see [16]. As a result, in our current work, we use the EMW to detect damage in wind turbine blades.

These articles [17,18] present a finite element analysis of detecting surface rust in steel rod models using point-source-generated ultrasonic waves, and they discuss the most recent advances in an ultrasonic structural health monitoring approach based on ultrasonic guided wave theory.

In [19], they employ an acoustic emission technique to evaluate an appealing damage detection strategy for structural composite components used in railway switches and crossings.

From the theory of inverse problems and experience in non-destructive testing, wave-based methods generally allow the detection of local changes and definitely ensure better accuracy than static approaches (see [20–22]). Therefore, the electromagnetic wave equations are considered in the current paper.

## 2. Materials and Methods

### 2.1. Preliminaries and Notations Related to Maxwell's Equations

Maxwell's equations describe how electric and magnetic fields propagate and interact and how they are affected by objects [23,24]. James Clerk Maxwell (1831–1879) formulated these equations based on a set of known experimental laws [25]:

$$\nabla \cdot \mathbf{D} = \rho_V, \qquad \text{Gauss's law,} \tag{1}$$

$$\nabla \cdot \mathbf{B} = 0, \qquad \text{Gauss's law for magnetism,} \tag{2}$$

$$\nabla \times \mathbf{E} = \frac{\partial \mathbf{B}}{\partial t}, \qquad \text{Faraday's law,} \tag{3}$$

$$\nabla \times \mathbf{H} = \frac{\partial \mathbf{D}}{\partial t} + \mathbf{J}. \qquad \text{Ampere's law.} \tag{4}$$

Here **D** is the electric flux density, $\rho_V$ is the electric charge density in Gauss's law, and **J** is the electric current density in Ampere's law. The electric flux density **D** is related to the electric field **E** by

$$\mathbf{D} = \varepsilon\mathbf{E}, \tag{5}$$

where $\varepsilon$ is the permittivity of the medium. The magnetic flux density **B** and the magnetic field **H** are related to each other by $\mathbf{B} = \mu\mathbf{H}$, where $\mu$ stands for the permeability. Consequently, we obtain

$$\nabla \cdot \mathbf{H} = 0. \tag{6}$$

Using the constitutive relations

$$\mathbf{D} = \varepsilon\mathbf{E}, \qquad \mathbf{B} = \mu\mathbf{H}, \qquad \mathbf{J} = \sigma\mathbf{E}, \tag{7}$$

where $\sigma$ is the electric resistance.

### 2.2. Electromagnetic Waves

One can rewrite Maxwell's equations only in terms of the electric and magnetic fields:

$$\nabla \cdot \mathbf{E} = \frac{\rho_V}{\varepsilon}, \tag{8}$$

$$\nabla \cdot \mathbf{H} = 0, \tag{9}$$

$$\nabla \times \mathbf{E} = -\mu\frac{\partial\mathbf{H}}{\partial t}, \tag{10}$$

$$\nabla \times \mathbf{H} = \varepsilon\frac{\partial\mathbf{E}}{\partial t} + \sigma\mathbf{E}. \tag{11}$$

By means of vector analysis, these equations can be transformed into two wave equations for the electric and magnetic fields, respectively.

$$\nabla^2\mathbf{E} - \mu\varepsilon\frac{\partial^2\mathbf{E}}{\partial t^2} = 0, \tag{12}$$

$$\nabla^2\mathbf{H} - \mu\varepsilon\frac{\partial^2\mathbf{H}}{\partial t^2} = 0, \tag{13}$$

where $c = \frac{1}{\sqrt{\mu\varepsilon}}$ is the speed of light (i.e., phase velocity) in a medium, and $\nabla^2$ is the Laplace operator.

### 2.3. Methods

We will use the electromagnetic wave equations in accordance with Maxwell's equations where **E** (electric field) and **H** (magnetic field) are coupled with each other. Electromagnetic waves are used to observe and quantify effects of cracks. We are interested in seeing if the presence of cracks can be identified using only one source of sent waves. By sending electromagnetic waves through a specimen and comparing the waves arriving at a receiver with the waves from the undamaged specimen, we will robustly check in the following sections whether the effects of existing cracks are visible in the signals or not.

## 3. Results

The objects are designed and simulated using the 3D electromagnetic field simulation (Empire XPU software), which is based on the powerful finite difference time domain (FDTD) method.

The first design with simple geometry is a rectangular cuboid shape. The second design, which is more complicated, is based on the simple shape of the blades. Last but not least, is the rotor blade design, which has a complex surface that resembles the actual form.

In these designs, two types of materials, conductor and dielectric, will be considered.

### 3.1. Using the EM Waves to Observe and See the Effects of Cracks in the Rectangular Cuboid Design

The very simple shape (rectangular cuboid) is seen in Figure 1.

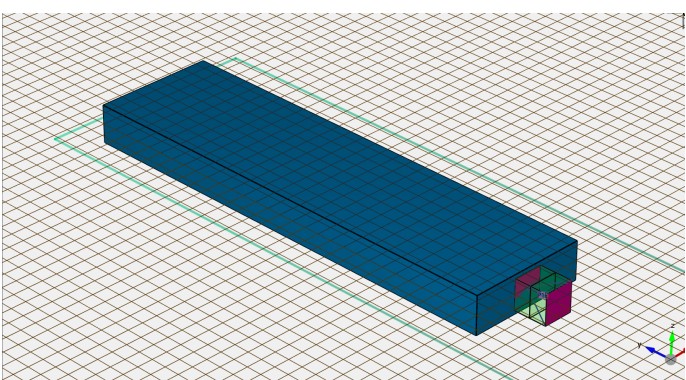

**Figure 1.** The rectangular cuboid object with a source.

In these designs, the source is attached to the surface of the object. We will use the electromagnetic wave equations to observe and detect any effects or cracks of two types. As in Figure 2, one of them has a crack, and the other has two cracks.

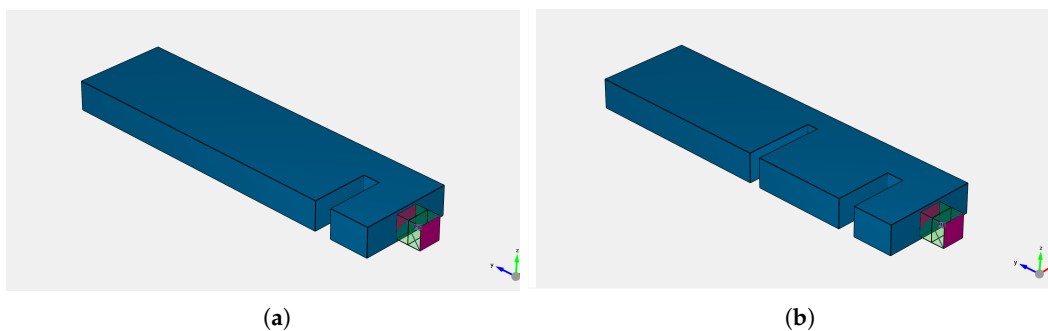

(**a**)          (**b**)

**Figure 2.** Objects with cracks: (**a**) one crack; (**b**) two cracks.

The source was designed for the band of operational frequencies $f_1 = 1$ GHz to $f_2 = 20$ GHz, and the target frequency is 13 GHz. For that bandwidth, the dimensions of the object should be proportional to the aforementioned frequency range.

The dimensions of the transmitter (the source) depend on the wavelength of the waves received, where those dimensions are within the wavelength of those waves:

$$\lambda = c/f, \tag{14}$$

where $\lambda$, $c$, and $f$ are the wavelength, the speed of light, and the frequency, respectively.

In the first case, we will use an aluminum-EC conductor with the following thermal properties: conductivity = 237 W/(m K), surface heat-sink coefficient = 20 W/(m$^2$ K), and surface radiation emission coefficient (rel.) = 1.

The dimensions of this object are 30 cm, 8 cm, and 2.5 cm for length, width, and height, respectively. In this case, the minimum distance between the first and second cracks is 8 cm.

In our simulation, we will generate the electric field by using the source and evaluate the data related to the reflected voltage on the object with cracks (one and with two cracks) to plot them and compare them with the reflected voltage on the object without cracks.

From this data, one can see and observe the existence of cracks as shown in Figure 3.

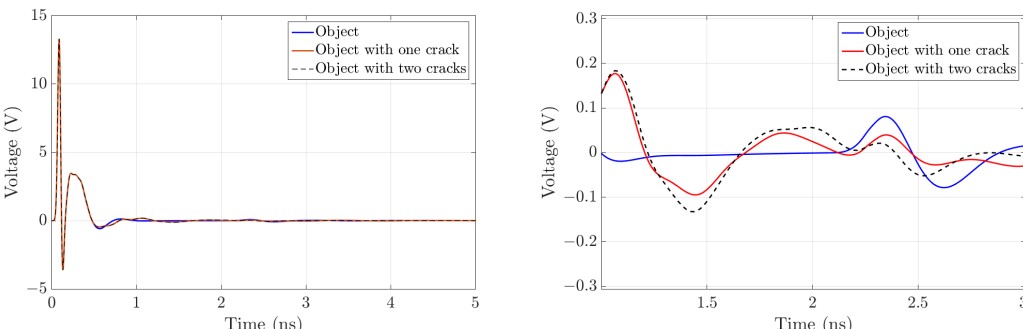

**Figure 3.** Comparison of reflected voltages in three different objects.

We can obtain more information from our simulation by computing the difference between the reflected voltage on the object with one or with two cracks and without cracks (no cracks), as seen in Figure 4. In addition, we can see some information regarding the location of the cracks.

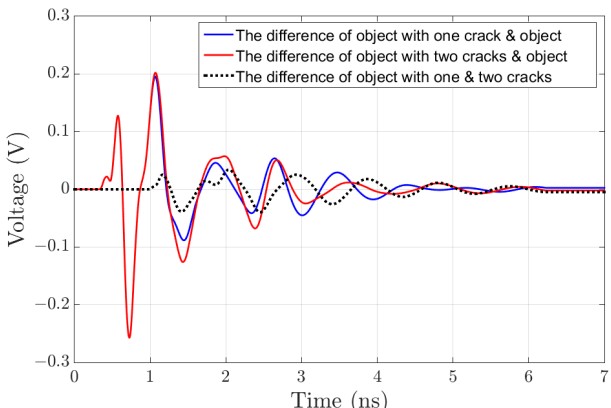

**Figure 4.** The difference in the amount of reflected voltage between the intact object and the cracked object.

Now we will switch to a dielectric material (epoxy-resin) with the following thermal properties: conductivity $= 0.2\,\text{W}/(\text{m K})$, surface heat-sink coefficient $= 1\,\text{W}/(\text{m}^2\,\text{K})$, and surface radiation emission coefficient (rel.) $= 1$. When the material of the rectangular cuboid object is changed to a dielectric material, we get better results, and one can see the changes in the signals due to cracks more clearly as shown in Figure 5.

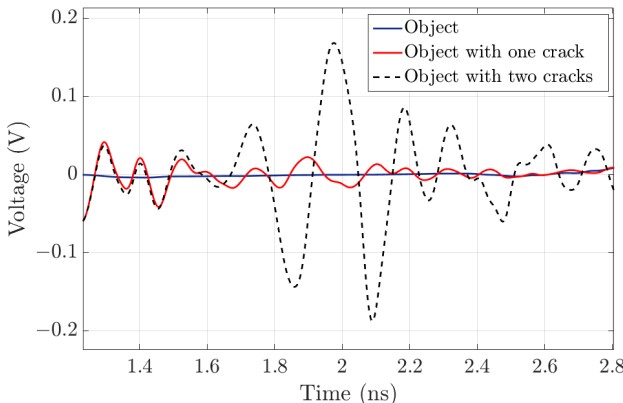

**Figure 5.** Comparison between the reflected voltage in the three different rectangular cuboid objects (epoxy-resin).

We utilized air as a backdrop material in all simulations (environments).

### 3.2. Using the EM Waves to Observe and See the Cracks in the Simple Blade Design

The final aim of this paper is to recognize cracks in the blades of the wind generator. For that purpose, firstly, we have designed the simple blade shape as shown in Figure 6 with the dimensions (length is 39.338 cm, thickness is 5 cm). Since the width of the simple blades is non-homogeneous, we measured the blade's widest width, which is 17.39 cm. Then, we will observe any effects or cracks by using the electromagnetic waves as explained in this section.

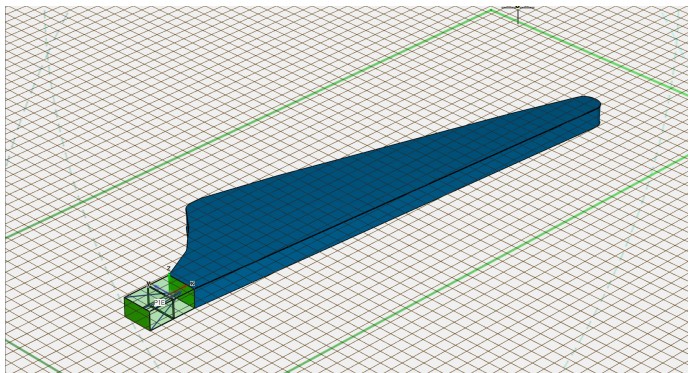

**Figure 6.** The simple blade design with a source.

In these designs, we will use the electromagnetic wave equations to observe and detect the effects of cracks in two cases. As seen in Figure 7, one of them has a crack, while the other has two cracks with a minimum distance of 22.5 cm between them.

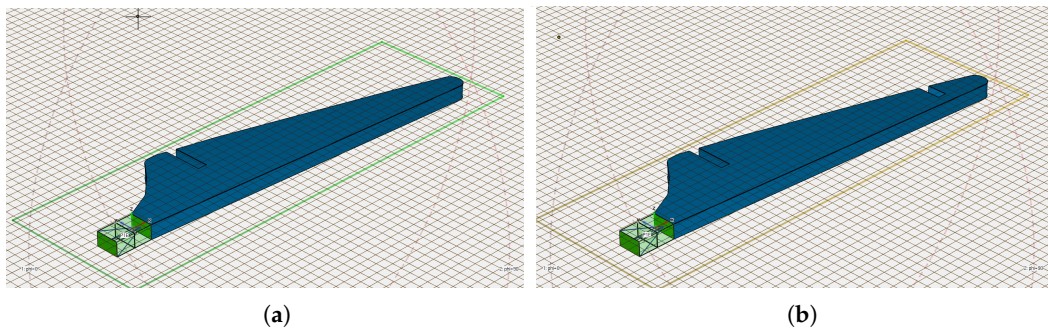

|(**a**)|(**b**)|

**Figure 7.** The simple blade design with cracks: (**a**) one crack; (**b**) two cracks.

Using a source to generate the electromagnetic waves, we can see and observe the cracks by comparing the waves (the reflected voltages) to the reference (simple blade without cracks), and we will get very good results, as shown in Figure 8.

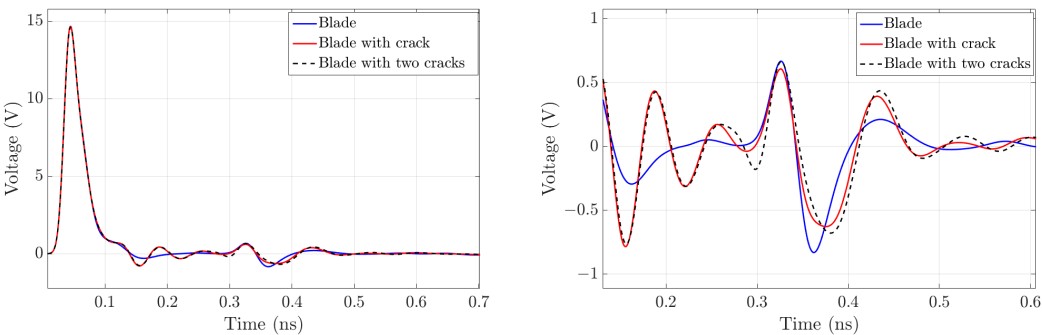

**Figure 8.** Comparison between the reflected voltage in the three different simple blades.

Due to the development of wind turbines (getting larger and taller), their recycling is still difficult and costly. Therefore, early removal should be avoided. Moreover, due to the high dynamic forces, structural changes (cracking) may occur in the majority of fragile materials even during run-time. Therefore, powerful and reliable testing tools (inspection tools) should be developed and applied.

The final goal related to this fact is that we should observe the cracks in the blades without moving them or even bringing them down, but rather by using sensor-supplied drones and measuring the electromagnetic wave equations.

To do this, we will need a source that generates electromagnetic waves without contacting the simple blades, as shown in Figure 9.

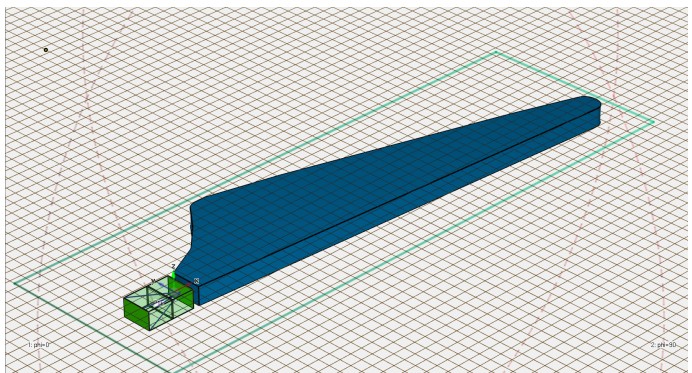

**Figure 9.** The source is not connected to the simple blade.

As illustrated in Figure 10, we can detect and notice the cracks even if the source is not contacting the simple blades.

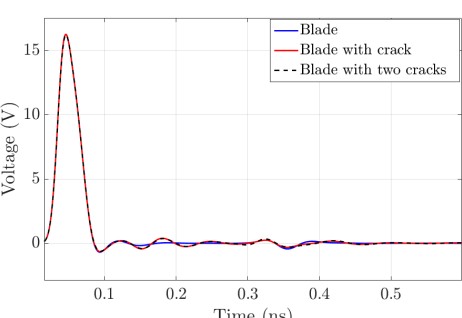
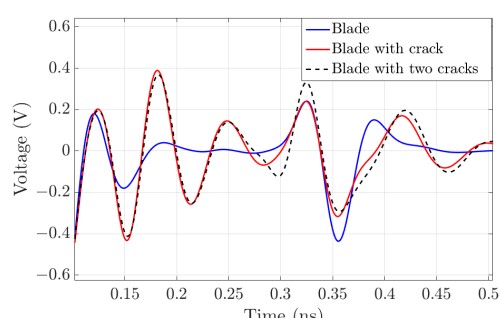

**Figure 10.** A comparison of the reflected voltage generated by a disconnected source in each of the three simple blades.

### 3.3. The Fast Fourier Transform (FFT)

The Fast Fourier Transform is a powerful tool for analyzing and measuring signals, and one can observe if there is any damage in the blade rotor. Therefore, we take the FFT of the signal (the reflected voltage) related to the simple blade without cracks and with one and two cracks.

We see a big change in the FFT plots for the simple blade without cracks and with one and with two cracks. We plot the results in Figure 11, and the effects of damage are visible.

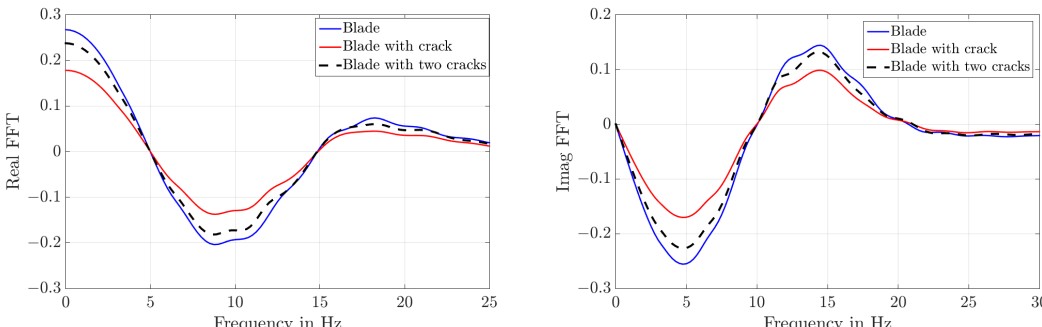

**Figure 11.** The real part and the imaginary part of the FFT for the signal are related to three different simple blades.

In addition, we checked the amplitude and phase of the spectra, and they show a clear result, as seen in Figure 12.

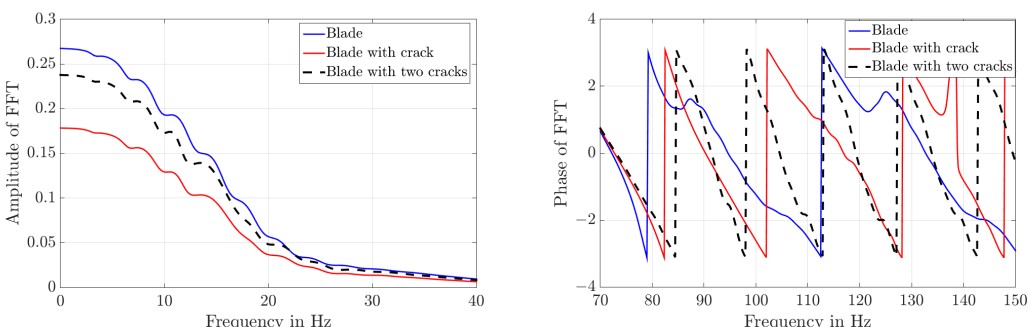

**Figure 12.** The amplitude and phase of the FFT for the signal are related to three different simple blades.

Now, the simple blade is bigger (the length is 6.24 m, the thickness is 24 cm, and the widest width is 1.4269 m). Figure 13 illustrates this. Here, we have chosen a special source (an anti-dipole Vivaldi antenna) to generate the electromagnetic waves.

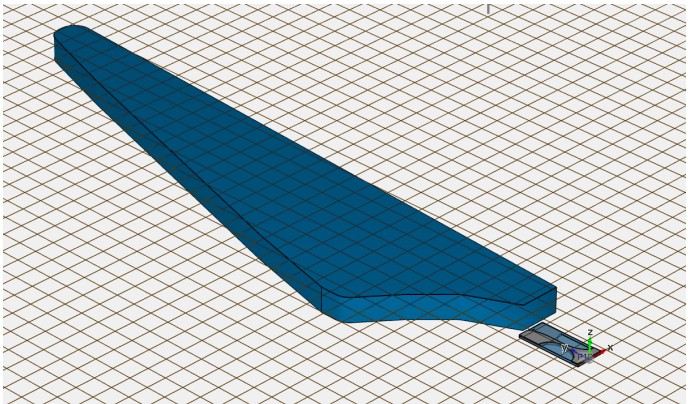

**Figure 13.** Simple large blade with an anti-dipole Vivaldi antenna.

This simple blade has cracks of various sizes and places, with a minimum distance of 4.2 m between them, as seen in Figure 14.

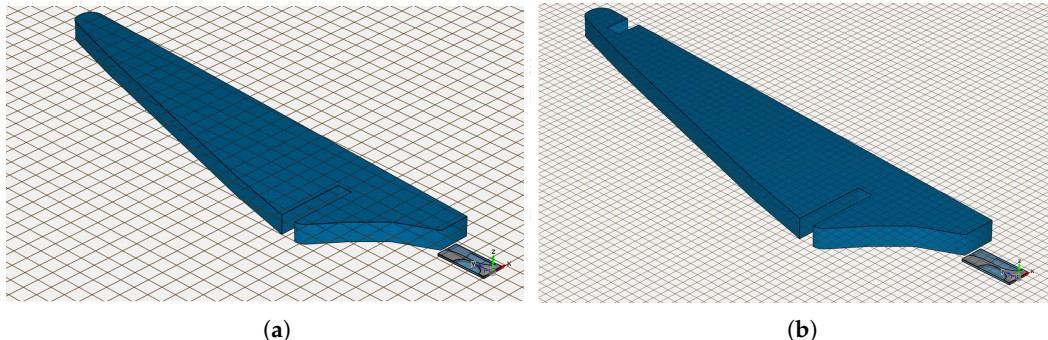

**Figure 14.** Simple large blades with different cracks and with an anti-dipole Vivaldi antenna: (**a**) one crack; (**b**) two cracks.

In the following Figure 15, we calculate the difference between the reflected voltage generated by the far anti-dipole Vivaldi antenna, firstly from the intact blade and the one-cracked blade and then between the intact blade and the two-cracked blade, respectively.

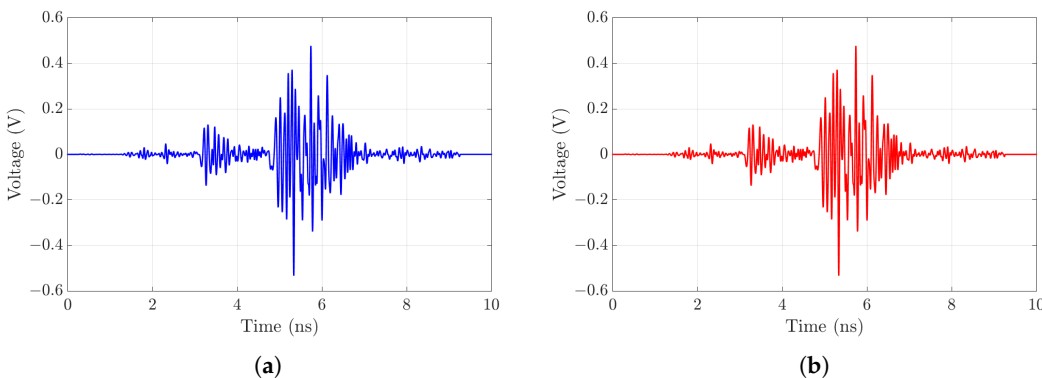

**Figure 15.** Comparison between the differences of the reflected voltage generated by the far anti-dipole Vivaldi antenna from simple large blades without cracks, with one crack, and with two cracks: (**a**) the distinctions between an intact and a cracked blade; (**b**) the distinctions between an intact blade and a two-cracked blade.

We observe that the existence of cracks can be clearly determined, but the responses to one and two cracks look very similar. To get a deeper understanding, we will now compare the responses for the cases of one and two cracks, as seen in Figure 16, respectively.

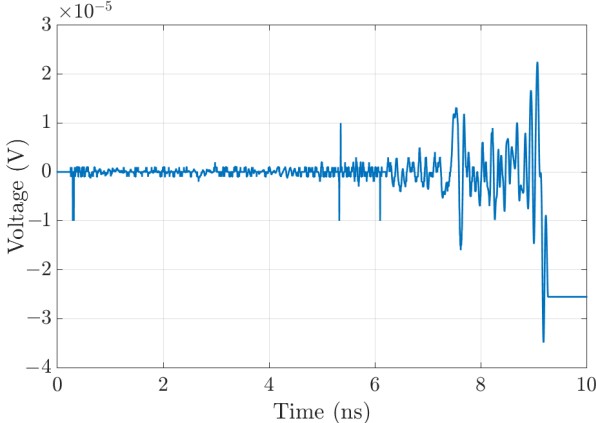

**Figure 16.** Measuring the reflected voltage generated by the far anti-dipole Vivaldi antenna from the simple large blade with one crack and two cracks.

One can see that these differences are of the order of $10^{-5}$. Compared with the original signals of order 1, we conclude that it is not possible with the described design of the experiment to distinguish one and two cracks, respectively. For this conclusion, we also have to take into account the inaccuracy of the measurements. Actually, we will change the material of this simple large blade to dielectric (epoxy-resin), generate the electromagnetic waves by using an anti-dipole Vivaldi antenna, and check again whether one can observe and see the cracks on it. The results of this investigation are shown in Figure 17. It is clear that we can observe the cracks in the simple large blade, which is made from epoxy-resin, by using the same method, and again we see that the existence of cracks is indicated, but the results for one and two cracks, respectively, are very similar.

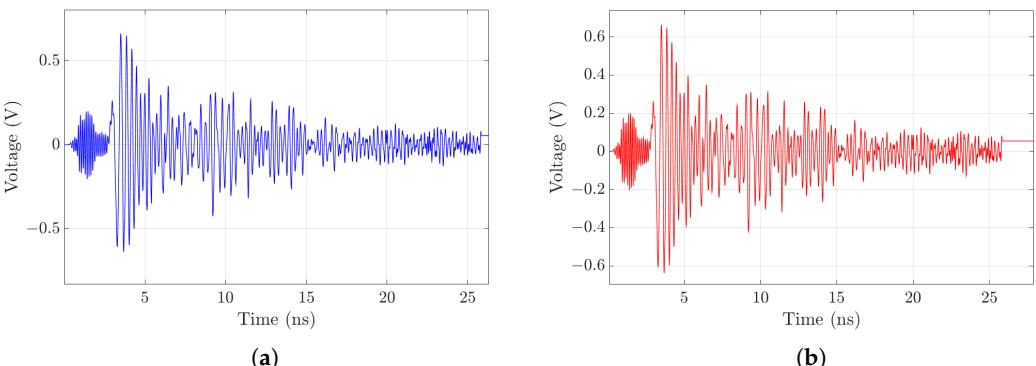

(**a**)　　　　　　　　　　　　　　　　　　　　　　(**b**)

**Figure 17.** Comparison between the differences of the reflected voltage generated by the far anti-dipole Vivaldi antenna from the epoxy-resin blade: (**a**) the distinctions between an intact and a cracked simple large blade; (**b**) the distinctions between an intact simple large blade and a two-cracked blade.

Analyzing the difference between the signals for one and two cracks, in Figure 18, we see again that the difference is too small to use it for distinguishing one or two cracks.

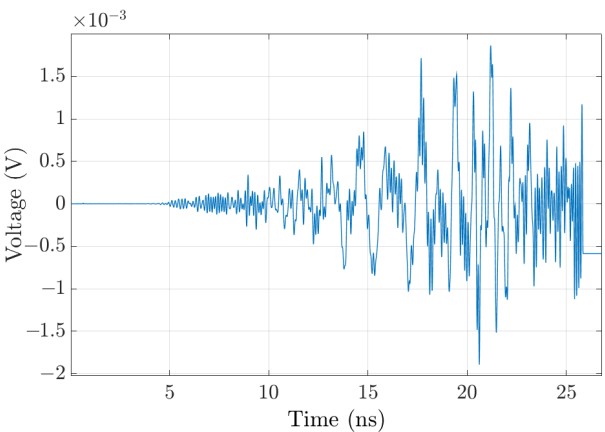

**Figure 18.** Measuring the reflected voltage generated by the far anti-dipole Vivaldi antenna on an epoxy-resin blade with one crack and with two cracks.

To finalize this paper, one should consider that the source will be in different locations, and the cracks will be hidden or inside the simple blade, as explained in the next section.

### 3.4. Source in Different Locations with Hidden Cracks (Internal Cracks)

In practice, determining the operating state of the rotor blades is more complicated because many cracks are invisible, and some of them are too small or narrow with non-uniform shapes. For these reasons, we studied two internal small cracks with a non-uniform shape (zigzag) inside the simple blade (length is 23.4 cm, thickness is 1 cm). As shown in

Figure 19, the length of the first and second zigzag cracks is 2.68 cm and 1.7 cm, respectively, and the minimum distance between the first and second cracks is 12.1 cm.

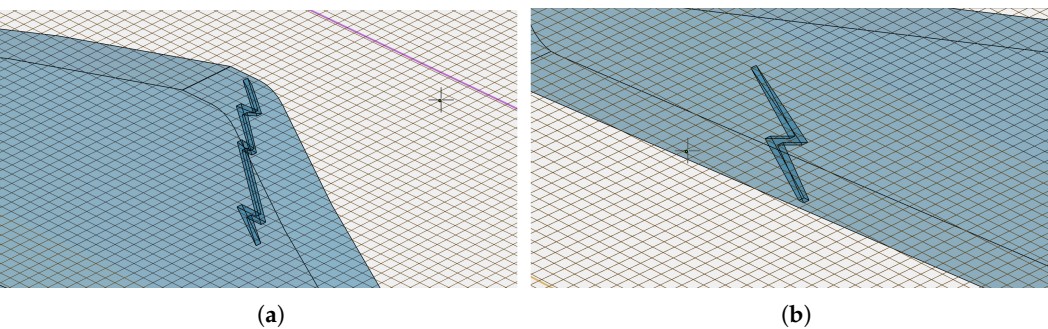

(**a**)  (**b**)

**Figure 19.** Narrow zigzag cracks that are not visible: (**a**) first crack; (**b**) second crack.

In addition, we will rotate the simple blade 90° in a clockwise direction, as seen in Figure 20.

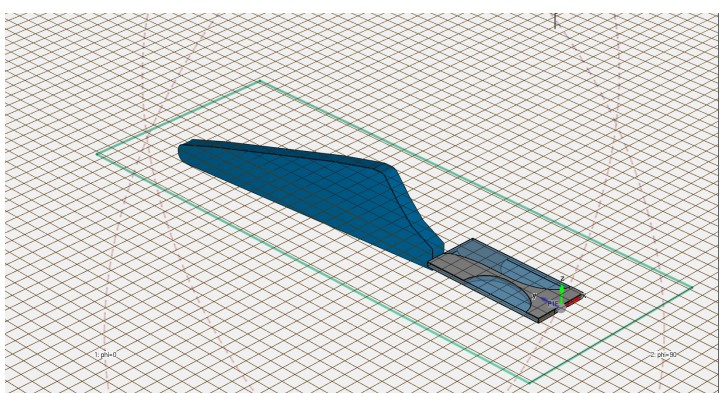

**Figure 20.** The simple blade rotates 90° in a clockwise direction.

We are repeating a similar procedure to observe the invisible cracks, and we could see the cracks by using the electromagnetic waves as shown in Figure 21.

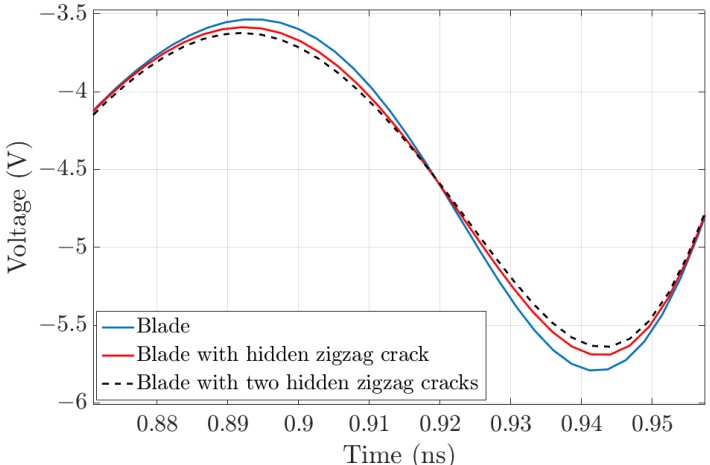

**Figure 21.** The difference in the amount of reflected voltage between the blades without cracks and those with one invisible zigzag crack and two cracks.

As mentioned earlier, the ultimate goal will be achieved through simulation using sensor-supplied drones and measuring the EM waves to observe and see if there are cracks

in the blades. Therefore, we will change the source location and perform our investigations using the same simple blade with the same invisible cracks.

So, we place the source on the backside of the simple blade, as shown in Figure 22.

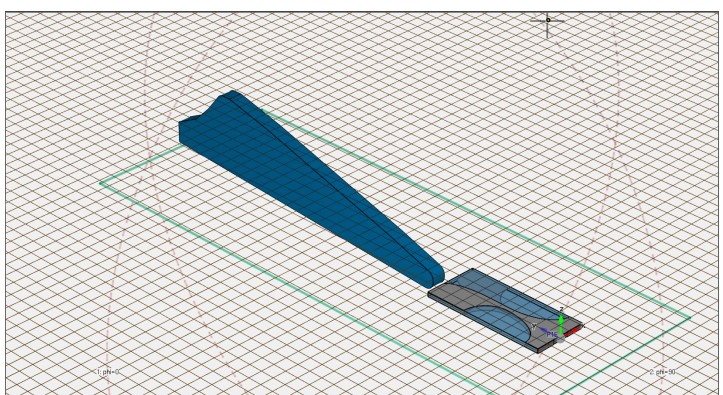

**Figure 22.** The source is at the backside of the simple blade.

Even when we change the location of the source, obviously, we can see and observe the invisible cracks in the simple blade more clearly compared with the previous location, as in Figure 23.

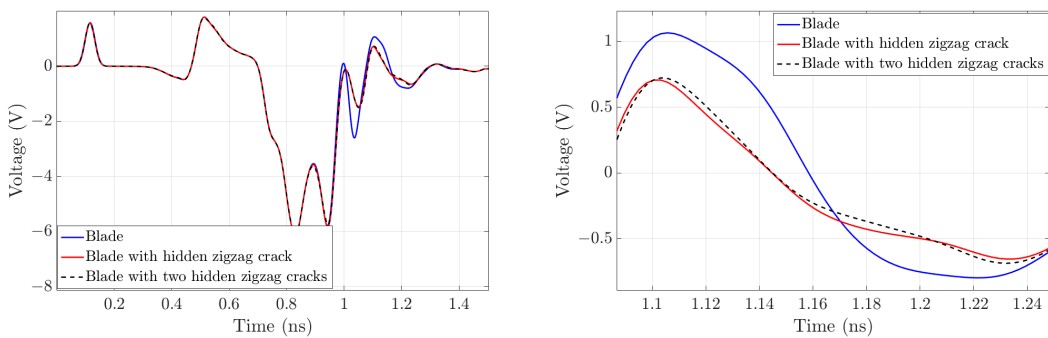

**Figure 23.** The difference in the amount of reflected voltage between the simple blade without cracks, with one invisible zigzag crack, and with two cracks.

We conclude that the change in the source location does not constitute a hindrance to crack detection, but in some cases, it can result in high accuracy, as shown in Figure 24.

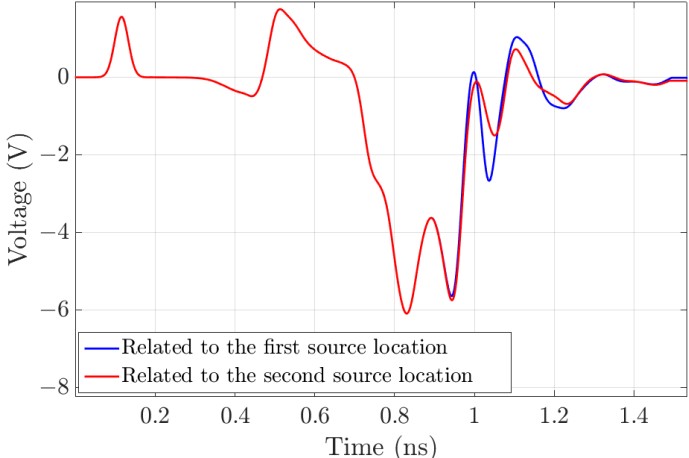

**Figure 24.** Thereflected voltage of the simple blade has two invisible zigzag cracks at two different source locations.

### 3.5. Using the EM Waves to Observe and See the Cracks in the Intricate Rotor Blade Design

Finally, we will consider the design of the rotor blade, which has a complicated surface with a leading edge, camber, twist, and various thickness distributions over the blade's length (here, the rotor blade's length is 38.1 cm), as shown in Figure 25.

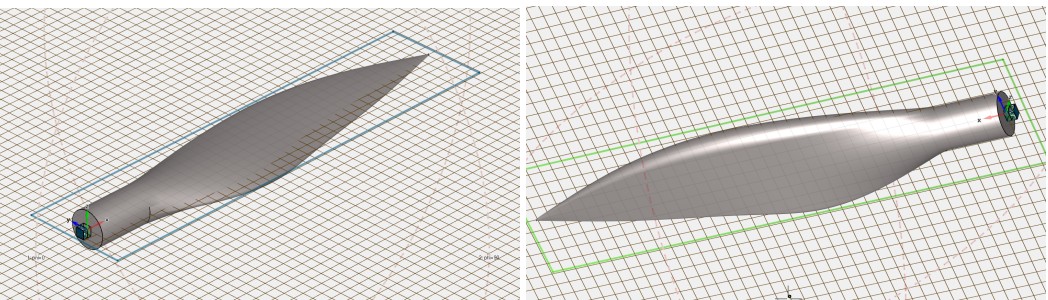

**Figure 25.** Rotor blade of a turbine with a source.

The rotor blade is made of a conductive material with the following thermal properties: conductivity = 100 W/(m K), surface heat-sink coefficient = 20 W/(m² K), and surface radiation emission coefficient (rel.) = 1.

Once more, we will use the electromagnetic wave equations to observe and detect the effects of cracks. Firstly, we check the undamaged rotor blade, and then we check the cracked rotor blade as seen in Figure 26, (the length of the crack is 13.3 mm and the width is 1 mm).

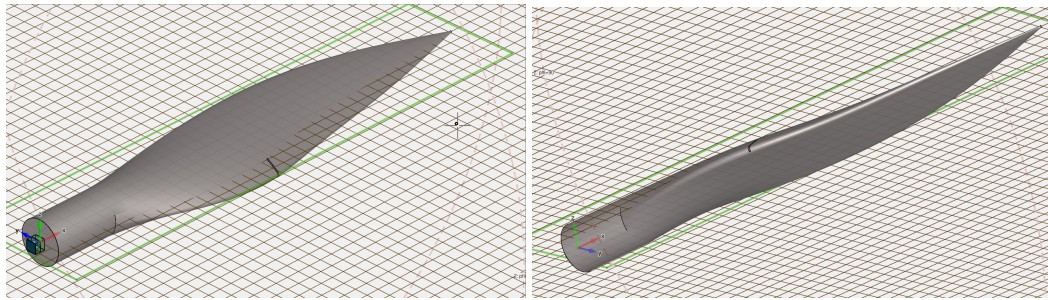

**Figure 26.** Rotor blade of a turbine with a crack.

We can still observe that the rotor blade has a crack even if the waves that are reflected from a flat surface are different from those that are reflected from the convex suction side and the concave pressure side of the blade, as shown in Figure 27.

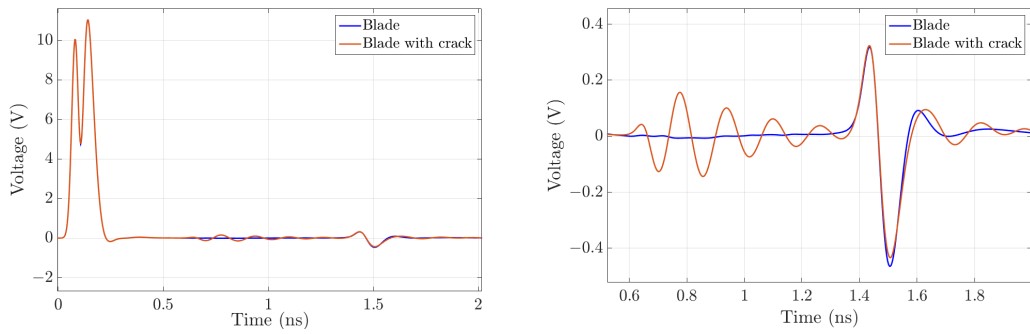

**Figure 27.** Comparison between the reflected voltage in the undamaged rotor blade and the cracked rotor blade.

Now, we use the same procedure to see the crack; in this case, the blade length is 1.143 m. As seen in Figure 28, it has a crack that is 0.192 cm wide and 0.417 cm long.

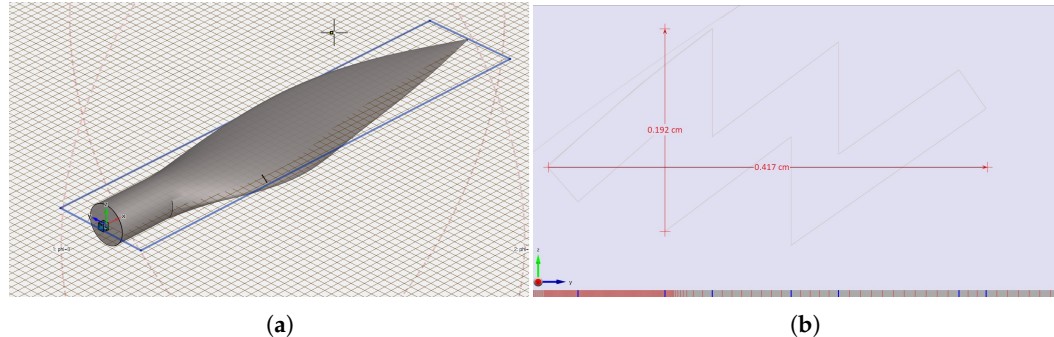

(**a**)                                            (**b**)

**Figure 28.** (**a**) The rotor blade with a crack; (**b**) the crack's shape.

By comparing the reflected voltage from the undamaged rotor blade and the rotor blade with a crack, as shown in Figure 29, one can determine if there is a crack.

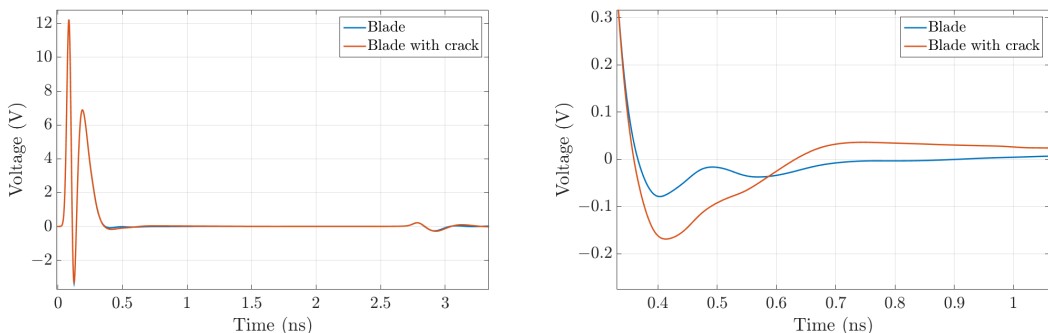

**Figure 29.** A comparison of the reflected voltage of the cracked and intact rotor blades.

We will take into account two minor hidden cracks in the rotor blade in the final simulation of this work. According to Figure 30, the first crack is 5.505 cm in length and 0.632 cm in width, while the second crack measures 2.875 cm in length and 0.50 cm in width.

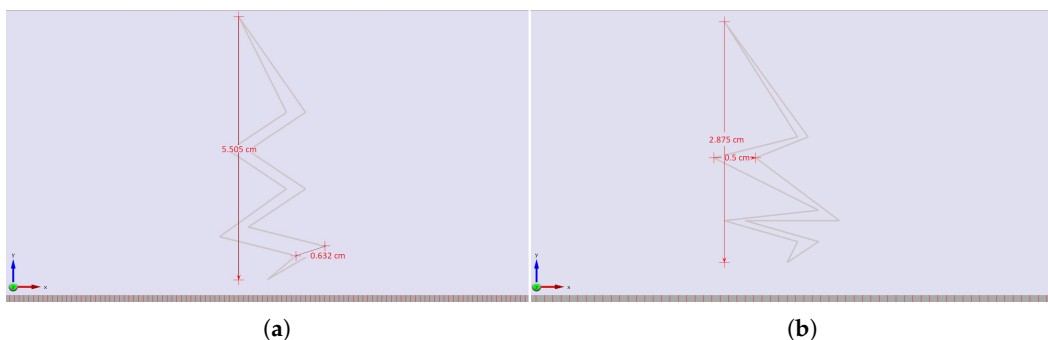

(**a**)                                            (**b**)

**Figure 30.** The shape of the cracks: (**a**) the first crack's shape; (**b**) the second crack's shape.

Figure 31 shows that the length of the rotor blade is 5.715 m, and the distance between the two cracks is 87.90 cm.

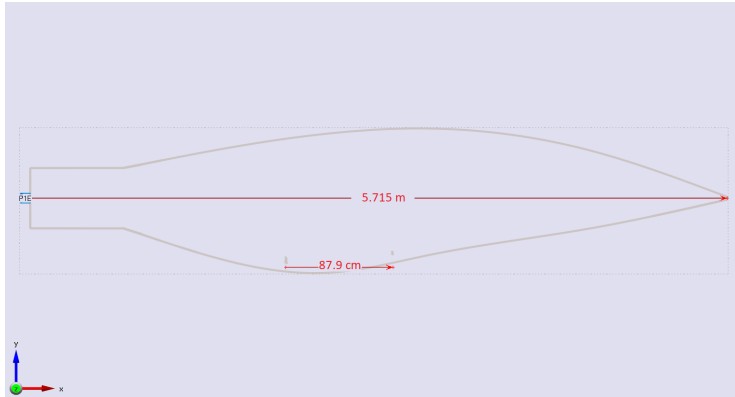

**Figure 31.** The rotor blade had two cracks.

Figure 32 illustrates how we may determine whether a crack exists even if the rotor blade, which has a complex surface, is large and the cracks are small by comparing the reflected voltage from the undamaged rotor blade and the rotor blade with one crack and with two cracks.

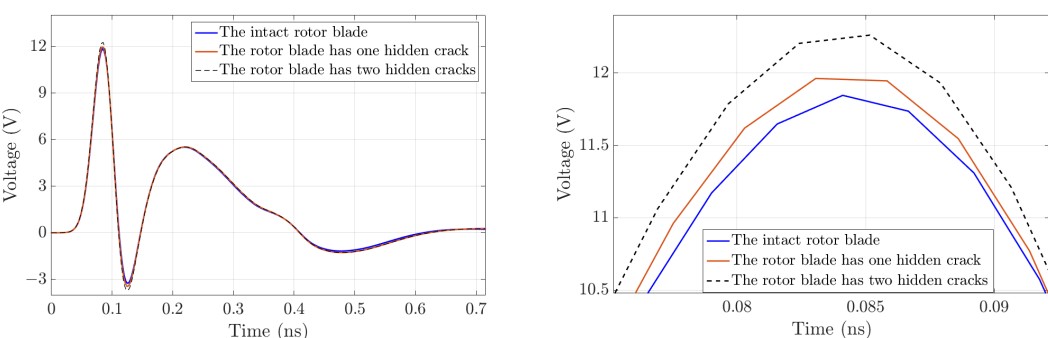

**Figure 32.** Comparison between the reflected voltage in the undamaged rotor blade and the hidden cracked rotor blades.

Now we will switch to a dielectric material with the following thermal properties: conductivity = 0.2 W/(m K), surface heat-sink coefficient = 20 W/(m² K), and surface radiation emission coefficient (rel.) = 1. We obtained better outcomes, and we can clearly identify the changes in the signals caused by the cracks, as shown in Figure 33, when the rotor blade material has changed to a dielectric material.

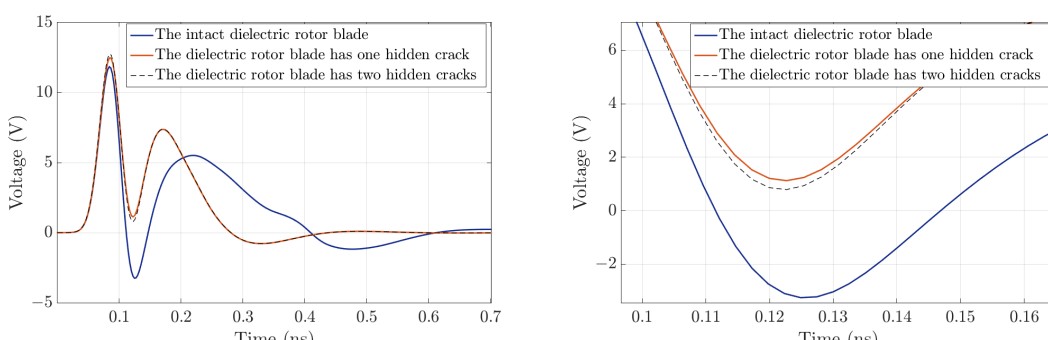

**Figure 33.** Comparison between the reflected voltage in the undamaged dielectric rotor blade and the cracked dielectric rotor blade.

## 4. Discussion

In this manuscript, we introduce preliminary results to identify the cracks in the rotor blades. The numerical simulations were performed with a small specimen for the

development of the approach as well as with bigger blades more closely related to the real world. In our model to detect cracks, the turbine is not operating.

For the calculations, the software EMPIRE XPU 8.01 was used to design and simulate the damage identification in wind turbines (rotor blades) by using the electromagnetic wave equations.

According to Maxwell's equations, the electromagnetic wave equations describe the interaction of electric and magnetic fields as the current is related to the magnetic field by Ampere's law, and the voltage is related to the electric field according to Maxwell's equations. In the above study, we used only the information related to the electric field (voltage) to see and determine the existence of cracks in the blades.

Using a source to generate the electromagnetic waves, we observed distortions of the signals due to the cracks. From these, we can conclude that the experimental setup seems useful for damage detection activities.

In most cases, the changes in the signals are sufficiently pronounced so that a damage-sensitive identifier can be applied. When time-series signals do not yet provide clear answers about the existence of a crack, the Fourier transform of the signals allows a more robust interpretation in the frequency domain.

In certain circumstances, when it comes to the direction, size, and location of the cracks, further data, e.g., on the magnetic field (current), should be used to produce superior outcomes. Further research is devoted to the full coupling of electromagnetic waves with elastic waves.

The simulations must be refined and extended to multiple measurements for the identification of the exact location and shape of the crack.

## 5. Conclusions

The present study shows that it is possible to observe and detect the existence of cracks in the blades by using the electromagnetic wave equations. Numerical simulations support the idea of proving the existence of cracks by only one sender and receiver. Comparisons between the undamaged and damaged blades, respectively, demonstrate that one can describe and see the effects of the cracks on easily accessible data. There are significant differences between the response signals in both cases. Thus, the described approach seems to be a good possibility for continuous monitoring in the sense of damage detection of the blades without disassembling them. Moreover, it has been demonstrated that the results do not seem to depend on the depth of the damage.

The use of an unmanned aerial vehicle (UAV) to identify cracks in the blades by emitting electromagnetic waves rather than moving or even bringing them down is the next step in the development of monitoring the cracks in the blades.

Finally, the examination of the rotor blades during operation, including the blade's flutter movement, temperature, and flow, are all aspects to consider for locating the cracks in future work.

**Author Contributions:** Conceptualization, Z.R.S.A.-Y., T.L. and K.G.; methodology, Z.R.S.A.-Y., T.L. and K.G.; software, Z.R.S.A.-Y. and H.M.M.; validation, T.L. and K.G.; formal analysis, Z.R.S.A.-Y., T.L. and K.G.; investigation, Z.R.S.A.-Y. and H.M.M.; resources, Z.R.S.A.-Y. and T.L.; data curation, Z.R.S.A.-Y. and H.M.M.; writing—original draft preparation, Z.R.S.A.-Y.; writing—review and editing, Z.R.S.A.-Y., T.L., K.G. and H.M.M.; visualization, Z.R.S.A.-Y.; supervision, T.L. and K.G.; project administration, T.L.; funding acquisition, Z.R.S.A.-Y. and T.L. All authors have read and agreed to the published version of the manuscript.

**Funding:** This research was funded by "Seed-Funding" (Bauhaus-University Weimar, Seed Financing Fund: Postdocs' funding line), https://www.uni-weimar.de/anschubfonds (accessed on 1 March 2020).

**Institutional Review Board Statement:** Not applicable.

**Informed Consent Statement:** Not applicable.

**Data Availability Statement:** Not applicable.

**Acknowledgments:** The authors highly acknowledge the support of Bauhaus-University Weimar while granting a seed-funding (Postdocs' funding line) to the first author (Zainab R. Al-Yasiri), within which this article was developed.

**Conflicts of Interest:** The authors declare no conflict of interest.

## Abbreviations

The following abbreviations are used in this manuscript:

| | |
|---|---|
| NDT | Non-Destructive Testing |
| EMW | Electromagnetic wave |
| GEW | Guided electromagnetic waves |
| EM | Electromagnetic |
| E | Electric field |
| H | Magnetic field |
| FDTD | Finite Difference Time Domain |
| FFT | Fast Fourier Transform |
| UAV | Unmanned Aerial Vehicle |

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
