# Peer review of "Damage Sensitive Signals for the Assessment of the Conditions of Wind Turbine Rotor Blades Using Electromagnetic Waves"

_infrastructures, doi:10.3390/infrastructures7080104_

Round 1
Reviewer 1 Report
Reviewer comments:
Good and valuable paper ready for publication after small corrections that authors must accomplish.
- Perhaps it would be more appropriate to specify the author and not just the bibliographic source number in the text;
- Figure 2 should be placed in the center of the page respecting the rule of the other figures;
- Pages 4 and 5 have a blank space at the base that should be used for good visibility of the work page;
- Figure 6 is a bit ambiguous regarding the description where it is said that a difference is made between two possible situations, but we have only one representation. Please pay attention and change if necessary;
- Figures 12, 13, 16, 18 on center of the page for an improved look of the page.

Author Response
Dear Sir,
We appreciate you reviewing our article and sending us some helpful feedback on our research.
I tried my best, together with my colleagues, to address the three reviewers' comments.
I apologise for the delay; it took a while to restart the simulation using the new rotor blade design, which has a complex surface with a leading edge, camber, twist, and various thickness distributions over the blade's length.
Best regards
Al-Yasiri, Zainab

Reviewer 2 Report
The paper ” Damage Sensitive Signals for the Assessment of the Conditions of Wind Turbine Rotor Blades using Electromagnetic Waves” reports a detailed description of an approach used for the determination of the conditions of wind turbine rotor blades through the use of electromagnetic waves. The topic of the manuscript is current and of considerable interest both the scientific community and designers but, it is opinion of this reviewer that some aspects of the manuscript were improved before publication in Infrastructures Journal:
MAJOR ISSUES
- the sections 4 and 5 appear to be insufficiently developed for a scientific publication. It is necessary to include a critical analysis of the results obtained and better highlight the limits of the proposed method. Furthermore, the conclusions should contain in detail the description of the innovative aspects of the research and future developments.
MINOR ISSUES:
- Figures 15, 17 and 19 are not indicated in the text;
- it is not clear what is the minimum distance between two consecutive cracks to be identified separately;
- In lines 18-20 consider as reported in 10.1016/j.istruc.2021.02.053;
- Better described the method in Section 2.
Author Response
Dear Sir,
We appreciate you reviewing our article and sending us many helpful comments on our research.
I tried my best, together with my colleagues, to address the MAJOR and MINOR ISSUES and to address the other reviewers' comments.
I apologise for the delay; it took a while to restart the simulation using the new rotor blade design, which has a complex surface with a leading edge, camber, twist, and various thickness distributions over the blade's length.
Best regards
Al-Yasiri, Zainab

Reviewer 3 Report
The authors have done some efforts to utilize Electromagnetic waves for crack detection in wind turbine blades. While the merit of the work is good, the execution needs many improvements. The model used in inaccurate and hence the proposed model cannot be used in the current form.
The manuscript overall needs rigorous revision regarding technical writing and English language proofreading. I overall recommend resubmitting this manuscript after major modifications have been made.
Major comments are as follows:
- The introduction part fell short to show the novelty of this work. Literature review was not sufficient to show relevant work, gap of knowledge, and the contribution of this work.
- The model used for wind turbine blade is totally inaccurate. Of course, some simplifications should be made, but showing the blade as a flat surface is far from correct. The blade has a complex surface, with a leading edge, camber, twist, and different thicknesses distributions along the blade length. The waves reflecting from a flat surface are not the same as the convex suction side, nor the concave pressure side of the blade.
- One more complication is the dynamic nature of turbine blades in operation. The blades are flexible allowing for vibrations and severe deflections (For large turbines, the blade tip can go as far as 4 meters away from the original position). How would your model be used to detect cracks while the turbine is operating?
- The authors keep referring to their work as "experimental" while this work is only a simulation. You should correct that. In the simulations adopted, the crack size is very large and yet the voltage difference is very low. How would you detect smaller cracks if the voltage difference is not observable? Even the zigzag crack with a width of 1 cm is a large crack and can be seen by bare eyes. How would your model handle crack size in order of milli or micrometers?
Minor comments:
- Overall proofreading by a native speaker who is familiar with the field is highly recommended.
- Some equations are not numbered, and variables are not defined in the correct order nor in the following paragraph.
- The first object simulated is better called (Rectangular cuboid/parallelepiped) instead of 3D rectangular.
- In Figure 6, the caption says, "difference between one and two cracks". However, the figure shows only one curve. Where is the difference?
Author Response
Dear Sir,
We appreciate you reviewing our article and sending us many helpful comments on our research.
I tried my best, together with my colleagues, to address your Major and Minor comments and to address the other reviewers' comments (major modifications have been made).
I apologise for the delay; it took a while to restart the simulation using the new rotor blade design, which has a complex surface with a leading edge, camber, twist, and various thickness distributions over the blade's length.
Thank you very much for your comments, which improved our search.
Best regards
Al-Yasiri, Zainab

Round 2
Reviewer 2 Report
The authors responded accurately to all of the this reviewer comments. The paper can be considered for the the publication in present form.
Author Response
Dear Sir,
Once again we appreciate your support and reviewing the article.
Best regards,
Zainab Al-Yasiri
Reviewer 3 Report
I would like to thank the authors for the extensive modifications made on the manuscript. A new complex blade has been made and simulations were performed over it, making it closer to the real-life problem. Still there are some modifications needed. However, I feel that is a good enough effort. You may mention that the manuscript introduces preliminary results, and add the shortcomings of the manuscript in the recommendations for future work, namely;
1- Examining blades in operation, including the flutter movement of the blade.
2- Using nonconductive materials to model the blade, since this is the reality of blades' materials (Mainly composites)
Author Response
Dear Sir,
Once again, we appreciate your support and reviewing the article.
According to your comments and suggestions,
In the discussion, we highlighted that this manuscript introduces preliminary findings to identify the rotor blade cracks.
Referring to your recommendations for future work:
We added in the conclusion that (the examination of the rotor blades during operation, including the blade's flutter movement, will be considered in future work).
As related to point 2, we already have a complex rotor blade design built of a dielectric material that is similar to the final complex rotor blade design with two hidden cracks, which has been added.
Best regards,
Zainab Al-Yasiri
